Psychosocial interventions for adults with visible differences: a systematic review

Norman Alyson 1 Alyson.norman@plymouth.ac.uk
Moss Timothy P. 2
1 School of Psychology, Plymouth University , Portland Square, Drake Circus, Plymouth , UK
2 Centre for Appearance Research, University of the West of England, Frenchay Campus , Bristol , UK
Moser Debra
Electronic publication date: 2015 Apr 2
Publication date: 2015
Volume: 3
Electronic Location ID: e870
Received 2014 Nov 24; Accepted 2015 Mar 10
Copyright: © 2015 Norman and Moss
Copyright year: 2015
Copyright holder: Norman and Moss
License: This is an open access article distributed under the terms of the Creative Commons Attribution License, which permits unrestricted use, distribution, reproduction and adaptation in any medium and for any purpose provided that it is properly attributed. For attribution, the original author(s), title, publication source (PeerJ) and either DOI or URL of the article must be cited.
License URL: https://creativecommons.org/licenses/by/4.0/

Keywords: Psychosocial, Narrative synthesis, Cognitive-behavioural therapy, Social skills training, Visible differences

Funding: The authors declare there was no funding for this work.

==============================
Background. Some individuals with visible differences have been found to experience psychosocial adjustment problems that can lead to social anxiety and isolation. Various models of psychosocial intervention have been used to reduce social anxiety and appearance related distress in this population. The objective of this review was to update a previous systematic review assessing the efficacy of psychosocial intervention programs for adults with visible differences. The original review (Bessell & Moss, 2007) identified 12 papers for inclusion.

Methods. A search protocol identified studies from 13 electronic journal databases. Methods: Studies were selected in accordance with pre-set inclusion criteria and relevant data were extracted.

Results. This update identified an additional four papers that met the inclusion criteria. Two papers provided very limited evidence for the efficacy of a combined cognitive-behavioural and social skills training approach. None of the papers provided sufficient evidence for the optimal duration, intensity or setting of psychosocial interventions for this population.

Discussion. The review concluded that a greater number of Randomised Controlled Trials and experimental studies were required to increase the methodological validity of intervention studies.

Introduction

The term visible difference refers to any kind of condition, whether congenital or acquired that can leave an individual with an altered appearance (e.g., skin conditions, burns, scarring or craniofacial abnormalities). Some individuals with visible differences have been found to experience psychosocial adjustment problems that can lead to social anxiety and isolation (Rumsey et al., 2004; Rumsey & Harcourt, 2012) and poor quality of life (Marcusson, Paulin & Ostrup, 2002). As such, appearance altering conditions present a clear challenge to a positive body image for those affected and have led to the development of numerous psychosocial intervention programs designed to address the psychological, as well as the physical needs and difficulties experienced by those with visible differences. The psychosocial difficulties experienced by some of those with visible differences include name calling, staring and unsolicited questioning about their appearance (Kleve & Robinson, 1999).

There are many different models that outline the difficulties experienced by some individuals with visible differences. These include the social anxiety model (Baumeister & Leary, 1995), Goffman’s (1968) model of stigma, social skills models (Bull & Rumsey, 1988) and models of body image disturbance (Cash, 2001). Baumeister & Leary’s (1995) model suggests that individuals with visible differences experience social anxiety at least in part because they are fearful of being rejected or excluded on the grounds of having an unusual or different appearance (Kent, 2000). Therefore, this model suggests that it is important to focus interventions on reducing social anxiety through exposure to social situations in order to promote positive adjustment amongst those with visible differences (Newell & Marks, 2000). Goffman’s (1968) stigma model fits in many ways with the social anxiety model, and states that having a different appearance is a characteristic that is “devalued” by society and as such those with a visible difference are more likely to be excluded or rejected, which suggests a very real reason for experiencing social anxiety.

Some research has suggested that those with visible differences can become preoccupied with their own appearance due to high levels of distress (Clarke, 1999). This preoccupation can make people seem distracted or lacking confidence when they are in public (Kent, 2000). Therefore, the social skills model suggests that many of the negative reactions that they experience from others are less to do with stigma, as Goffman’s (1968) model would suggest, but more a reaction to the poorer social skills that the person with the visible differences is exhibiting (Bull & Rumsey, 1988). These two models do not necessarily have to be mutually exclusive. The reality of the situation for many people with visible differences is indeed that they experience some level of rejection and exclusion from others, but in some cases this effect is exacerbated by the poor social skills that they have developed (Kent, 2000). Therefore, focusing on improving social skills is a key focus for intervention models (Rumsey, Robinson & Partridge, 1993).

Finally, the body image disturbance model (Cash, 1996) suggests that in the case of visible difference, the individuals may experience dissatisfaction with their body image because they do not conform to the cultural norms of attractiveness that their society imposes. This social pressure to look a certain way, alongside a more personal form of stigma, where they themselves feel they should look “normal,” can lead to high levels of body image disturbance, which is associated with poorer adjustment (Altabe & Thompson, 1996). This model suggests that interventions should focus specifically on addressing the way individuals feel about their appearance and the negative assumptions they make about the importance of appearance.

The reality is that no one model completely explains the experience of living with a visible difference. Kent (2000) recommended an integrated model that addresses body image dissatisfaction and the negative assumptions associated with appearance concerns. He also suggested that it is important to target social anxiety with exposure therapy (introducing people to feared social situations). However, as there is a very real tendency for individuals to experience negative responses from others, it is important to boost social skills too, in order to provide individuals with the techniques that they will need to deal with these responses. Both social skills training (SST) and cognitive behavioural therapy (CBT) are common intervention types for adults with visible differences.

Although these intervention techniques for people with a visible difference are used, there is still a significant lack of evidence pertaining to the efficacy of these different psychosocial techniques. A systematic review conducted by Bessell & Moss (2007) found little to no evidence to support any particular intervention model, due to methodological constraints associated with the included studies.

A recent systematic review conducted by Muftin & Thompson (2013) looked at self-help psychosocial interventions for individuals with visible differences. Whilst this is an important update, the review does not incorporate all forms of psychosocial intervention, only those administered in a self-help format. Additionally, the interventions did not necessarily measure body image dissatisfaction or appearance-related distress which is a key variable that can identify level of distress in the study of visible difference (Clarke et al., 2008). Therefore the review does not help to answer fundamental questions raised by the original review regarding method of delivery (Bessell & Moss, 2007). Furthermore, it is imperative that psychosocial interventions are employed using an evidence-based practice approach (Anderson, 2006). For that reason, it is important that the original review be updated to ensure an accurate evidence base for psychosocial interventions for this population. It is therefore the belief of the current authors that this update is both needed and timely.

Objectives

The aim of the present study is to assess the evidence-base for psychosocial intervention programs for adults with visible differences with a view to reducing appearance-related distress, body image dissatisfaction. This will be done through an update of the existing systematic review (Bessell & Moss, 2007) from 2006 (the date of the last search) to the present day. Where appropriate, meta-analysis was used to synthesise findings across papers. The overall intention of this study was to identify the evidence base for the therapeutic technique employed, as well as the method of delivery and optimal length and intensity of therapy.

Methods

Study selection

The search aimed to identify all studies relating to psychosocial interventions for adults with visible differences from January 2006 (six months prior to the original search in (Bessell & Moss, 2007)) to 12th May 2014. An extensive search strategy was used to search 13 databases, including Medline, embase, psychinfo, and Cochrane central register of Controlled trials (CENTRAL) (See Appendix for full search strategy). This was compiled by a library technician based on an exhaustive list of appearance altering conditions and types of psychosocial intervention. No language restrictions were applied. In addition, websites including National Institute of Clinical Excellence (NICE) and the metaRegister of Controlled Trials (mRCT) were searched and reference lists of included papers. Search criteria were adapted to suit the search terms of each individual database.

Inclusion criteria

Study design. No exclusions were applied based on study design with all study designs being included in the review. Case studies with less than five participants in each group were excluded.

Population. Adults with noticeable visible differences, e.g., disfigurements of face, neck and hands. This included a wide range of different conditions from congenital skin conditions and abnormalities to cancer patients, or those with scars resulting from injury. All client groups were over the age of 16. Both males and females of any ethnicity or race were included.

Interventions. These included CBT, SST and more traditional forms of psychotherapy all delivered either alone or as part of a package of care. The interventions had to include some element specifically designed to target appearance concerns.

Comparators. The comparators used in this review were current standard treatments including standard therapist-led CBT for the treatment of anxiety or depression, non-directive counselling, primary care counselling, routine management (drug treatments for anxiety or depression) and no treatment.

Outcomes. The primary outcome measure was any measure of appearance related distress (e.g., body image concerns, body image quality of life etc.). Only studies with this primary outcome measure were included in the review. 1 Secondary measures included measure of anxiety and depression and general improvements in psychological symptoms, interpersonal and social functioning, satisfaction and preference, site of delivery and acceptability of treatment.

Exclusion criteria

Any treatment designed to treat body dysmorphic disorder or eating disorders such as bulimia nervosa or anorexia nervosa were excluded. It was also decided to exclude any visible differences that were not considered to be commonly on display (such as breast reconstruction, abdominal injury). These types of conditions do fall within the remit of visible differences, but it was considered that the intervention needs of individuals with “hidden” differences might be different to those with normally visible differences, meaning that different intervention techniques may be appropriate. For example, social skills models are often used to help individuals manage questioning around appearance (Kent, 2000). This particular model may be less helpful to those with hidden differences who do not experience unsolicited questioning.

Analysis

The authors used a qualitative approach to synthesise data across studies (Dixon-Woods et al., 2005) and focused on three main areas: information pertaining to theoretical or therapeutic perspective, method of delivery (setting, person delivering the intervention) and timing of the intervention (intensity and frequency of the intervention).

Meta-analysis of trials only

Outcome measures. Primary and secondary outcome measures of psychosocial adjustment were extracted (e.g., preoccupation with appearance, anxiety, depression, confidence, quality of life, social integration).

Effect sizes. Standard mean differences (SMDs) and/or effect sizes together with 95% confidence intervals (CIs) were extracted for continuous outcomes and odds ratios (ORs) together with 95% CIs were extracted for dichotomous outcomes. Meta analyses were only conducted on multiple randomised controlled trials (RCTs) of similar interventions to allow appropriate data pooling.

Assessment of risk of bias

Three reviewers (AN, AM & JG) independently assessed trials using the Cochrane Risk of Bias tool (Higgins & Green, 2011). In the case of observational studies, two reviewers (AN & JG) used the RAMbo assessment tool (Chen & Wang, 2009) to assess the quality of randomization (R), whether missing data was accounted for (A) and whether the type of measurement was appropriate (M).

Results

The search results identified 13,837 possible studies since the previous review was conducted. After removal of duplicates, 3,539 studies were identified for further investigation. Of these 3,468 studies were identified as not relevant for inclusion within the review and were discarded on the basis of titles and abstracts independently by at least two reviewers (AN, AM & JG). This left 71 studies to assess for inclusion (see Fig. 1).

Figure 1 Flow diagram of search results.

Of the 71 papers identified for possible inclusion, four met the inclusion criteria on closer inspection by three reviewers (AN, TM & AM). Sixty-seven studies were excluded. Reasons for exclusion included studies that did not assess an intervention targeting appearance or related psychosocial distress (27 studies), those that did not assess an intervention (five studies), case studies with less than 5 participants in each group (13 studies), descriptive articles or review papers (14 papers), those with no primary outcome measure of appearance-related distress or body image concern (6 studies) and two which met the inclusion criteria, but not enough data was present in the abstracts to include within the review (authors were contacted for full papers but were not supplied).

Risk of bias assessment

Two papers (Srivastava & Chaudhury, 2014; Bessell et al., 2012) were assessed using the Cochrane risk of bias assessment tool which is suitable for assessing RCTs (Higgins & Green, 2011). The Bessell et al. (2012) paper was assessed for risk of bias by two researchers independent of the paper’s authors (AM & JG), as two of the authors were also the authors of this review.

Risk of bias assessment: Of the two papers, one (Bessell et al., 2012) was found to be of low risk of bias with regards to randomization sequence and allocation concealment (see Table 1). Only one paper was found to have low risk of bias for blinding of outcome assessor (Bessell et al., 2012). All rates of attrition were adequately documented in the papers. Srivastava & Chaudhury (2014) did not report any attrition rates throughout the study period. All outcomes reported in the studies were reported in the results.

Table 1 Risk of bias in RCTs.

Study	Study design	Sequence generation	Allocation concealment	Method of blinding of outcome assessor	Completeness of outcome data	Reporting of outcome data	
Bessell et al. (2012)	RCT	Low	Low	Low	Low	Low	
Srivastava & Chaudhury (2014)	RCT	Unclear	Unclear	Unclear	Low	Low	
Notes.

RCT Randomised controlled trial

Low low risk of bias

High high risk of bias

unclear information in the paper not sufficient to assess risk of bias

RAMbo assessment. Two papers (Jolly et al., 2010; Semple et al., 2009) were assessed using the RAMbo technique for observational studies (see Table 2). Jolly et al. (2010) did not report using a randomisation procedure, so was rated as unclear, whilst (Semple et al., 2009) did not use a randomisation technique so was rated at high risk of bias. Semple et al. (2009) was rated at low risk of bias for attrition and measurement, whereas Jolly et al. (2010) was rated as unclear as multiple abstract publications of this study refer to different numbers of participants. The study was also rated unclear for measurement as results for the anxiety outcome measure were not reported.

Table 2 Risk of bias observational studies.

Study	Study design	Randomisation procedure	Attrition	Measurement	
Semple et al. (2009)	Observational	High	Low	Low	
Jolly et al. (2010)	Observational	Unclear	Unclear	Unclear	
Notes.

Low low risk of bias

High high risk of bias

unclear information in the paper not sufficient to assess risk of bias

Effects of interventions: therapeutic approach

Cognitive-Behavioural Therapy. Jolly et al. (2010) assessed the efficacy of an individual CBT program for patients with lupus. The intervention focused on body image education, self-esteem, anxiety and depression and also contained cosmetic training. The study employed 15 women with lupus (10 treatment and 5 controls) through a clinic in the United States. The mean ages of the participants in the treatment and control groups were 43.6 years and 39.3 years, respectively. Outcome measures included Multi-Dimensional Body Relations Satisfaction – Appearance Scale (MBRSQ-AS), Situational Inventory of Body Image Dysphoria (SIBID-SF), Body Image in Lupus Screen (BILS) and Anxiety and Lupus PRO (Table 3).

Table 3 Characteristics of included studies.

Study	N	Location	Population	Age	Study design	Intervention	Comparator intervention	Setting	Facilitator	Intensity	Duration	Follow-up	
Srivastava & Chaudhury (2014)	90a	India	Adults with amputations	22–52 yrs	RCT	Person-centred counselling	Treatment as usual	Not stated	Psychiatric nurse	6 weekly sessions	Not stated	No follow up reported	
(Bessell et al., 2012)	83 (49 f)	United Kingdom	Adults with any visible difference	18 +	RCT	CBT/SST	No treatment control	Clinic	Therapist/ self help	8 weekly sessions	1 h	6 month post-intervention	
Jolly et al. (2010)	15 (15 f)	United States	Women with Lupus	18 +	CT	CBT/cosmetic training	No treatment control	Clinic	Therapist	10 weekly sessions	1.75 h	Week 18 & 24 post intervention	
(Semple et al., 2009)	54 (28 F)	United Kingdom	Head and neck cancer patients	31–75 yrs	CT	CBT/SST	Usual care	Home	Clinical nurse specialist	2–6 fortnightly sessions	90 min	3-month follow-up	
Notes.

a Not all studies reported gender. Figures are provided where reported

The previous review by Bessell & Moss (2007) did not include meta-analyses. The authors of the current review revisited the data from previous papers with a view to conducting meta-analyses on any studies that consisted of randomised trials. Two of the original papers met this criterion (Papadopoulos, Walker & Anthis, 2004; Newell & Clarke, 2000). The Newell & Clarke (2000) paper did not contain sufficient detail for meta-analysis. No other CBT studies consisted of randomised trials, so it was not possible to conduct a meta-analysis on this intervention type. Overall, the review concluded there was very limited evidence for the efficacy of CBT for adults with visible differences.

Combined CBT and SST. Bessell et al. (2012) assessed the efficacy of two psychosocial interventions against a no-treatment control. The first intervention consisted of a face-to-face CBT/SST intervention, whilst the second was an online delivery of the same intervention model. The study employed 83 individuals with varying visible differences recruited through charity organizations, the Royal Free Hospital, London outpatient plastic surgery clinics and general advertising. Participants (34 male, 49 female) were over 18 years of age, with a mean age of 45 years (see Table 2 for study information). Outcome measures used included the Hospital Anxiety and Depressions Scales (HADs), the Derriford Appearance Scale-24 (DAS-24), and the Body Image Quality of life Inventory (BIQLI).

Semple et al. (2009) assessed the efficacy of an individual CBT/SST program for patients with head and neck cancer. The intervention focused on a series of specific areas including anxiety, depression, fatigue, appearance and stress. The study employed 54 patients with head and neck cancer recruited through the Regional head and Neck service in Northern Ireland. Participants (40 males, 14 females) were 31 to 75 + years of age. Outcome measures included the HADs, the Work and Social Adjustment (WASA) scale and a health-related quality of life measure (University of Washington quality of life scale version 4) which contained a measure of appearance-related distress.

The Semple et al. (2009) paper did not contain sufficient detail to allow a meta-analysis to be conducted. No other CBT studies consisted of randomised trials, so it was not possible to conduct a meta-analysis on this intervention type. Overall, the review found only very limited evidence for the efficacy of a combined CBT and SST approach for adults with visible differences.

Person-centred: Srivastava & Chaudhury (2014) compared treatment as usual (one counselling session; 83 participants) against a six session psychotherapeutic program (90 participants). Participants were aged 22–52 years of age with a mean age of 30.05. All patients had experienced amputation. Intervention consisted of six session based on reassurance, ventilation of emotions, acceptance of self, therapeutic milieu and reintegration.

A study previously cited in the Bessell & Moss (2007) review also assessed the efficacy of a person-centred approach (Papadopoulos, Walker & Anthis, 2004). However, this study did not contain enough information to allow a meta-analysis to be conducted. Overall, this review has found little evidence for the use of the person-centred approach to therapy.

Effects of interventions: method of delivery

Self-help: One of the included studies assessed the efficacy of self-help interventions. The Bessell et al. (2012) paper compared face-to-face delivery of a CBT intervention against an online delivery with minimal facilitation from an assistant psychologist or counsellor.

Face-to-face individual: All four studies assessed the efficacy of individual CBT-based interventions. The Bessell et al. (2012) paper also assessed the efficacy of a face-to-face delivery of a CBT/SST intervention administered by a trained counsellor or an assistant psychologist. The Semple et al. (2009) paper assessed a face-to-face CBT/SST intervention administered by a trained clinical nurse specialist. Jolly et al. (2010) assessed the efficacy of individual CBT-based support for women with lupus. Srivastava & Chaudhury (2014) assessed the efficacy of individual psychotherapy delivered by a psychiatric nurse for individuals with amputations.

Due to the differences in methodological design, it was difficult to draw any firm conclusions about the optimal delivery of psychosocial interventions. Therefore, the review cannot recommend whether any particular individuals should be responsible for delivering these psychosocial interventions.

Effects of interventions: timing of intervention

This review attempted to identify the optimal duration and intensity of intervention. The studies included within this review varied in duration from two sessions (Semple et al., 2009), through to 10 sessions (Jolly et al., 2010). Full details of intervention duration can be found in Table 3. The intensity of the interventions consisted of weekly (Srivastava & Chaudhury, 2014; Bessell et al., 2012) or fortnightly sessions (Semple et al., 2009). Sessions were between one and two hours in length (see Table 3 for full details of intensity).

Due to the differing intensity and duration across the studies, it is difficult to draw any firm conclusions regarding the optimal length and intensity of therapy. However, most studies opted for between 6 and 10 sessions administered weekly for 1–1.5 h. Therefore, it would seem reasonable to conclude that this is the minimum intensity and duration required to lead to clinically significant changes in appearance-related distress and anxiety. This also matches recommendations for the minimum intensity of therapies in the general population (Roth & Fonagy, 2005).

Effects of interventions: participant acceptability

As well as assessing efficacy of interventions, it is important that trials of interventions also measure patient acceptability. One paper reported on overall acceptability (Bessell et al., 2012; Newell & Clarke, 2000). The Bessell et al. (2012) provided information about overall acceptability, as well as ratings of usefulness and satisfaction for both the face-to-face and computer-based intervention. Users of the face-to-face intervention gave it an average usefulness rating of 8.23 out 10 and a satisfaction rating of 8 out 10. The computer intervention was given ratings of 8.79 and 8.38 out of 10, respectively. Overall acceptability for the face-to-face intervention was 51.89 out of 60 and 52.7 out 60 for the computer intervention. The original Bessell & Moss (2007) review also included a study by Newell & Clarke (2000) which measured patient acceptability (not included in the previous review). (Newell & Clarke, 2000) paper found that 68.75% found the leaflet useful. Only 9.38% rated the booklet as unhelpful. These papers suggest that the CBT or combine CBT and SST approach may be viewed as acceptable by adults with visible differences.

Discussion

Main findings

The strength of the evidence to support the efficacy of the existing interventions from this narrative synthesis is generally poor. The methodological quality of the included studies was limited and small intervention effect sizes were observed. The studies looked at differing interventions making judgments about consistency across studies difficult because each study used different intervention settings, e.g., group, self-help or face-to-face and paradigms, e.g., CBT, SST or person-centred. There is some very limited evidence to support the efficacy of a combined CBT and SST approach, but this is far from conclusive as it is based on a combined sample size of 137 participants.

The length of intervention required was unclear with studies ranging from six to 10 sessions. No firm conclusions can be made regarding the optimum therapy time required to reduce psychosocial difficulties, or the most appropriate setting for these interventions. Neither can conclusions be drawn about the level of therapist contact or expertise required to produce optimum results. Due to the wide-ranging use of therapeutic paradigms of each intervention, it was not possible to draw any firm conclusions regarding the acceptable content of psychosocial interventions for the visibly different population, or the adequate implementation of these interventions. The participant populations were also varied in terms of conditions and symptom severity. Further studies need to be conducted to establish which interventions are most effective for specific sub-populations.

Interpretation of findings in relation to previously published work

The findings of this review were no different to the conclusions of the original review (Bessell & Moss, 2007), which made recommendations for a greater number of future studies, including more RCTs and experimental studies. Furthermore, the need for greater methodological vigour was highlighted with regards to ITT analyses, greater detail pertaining to attrition characteristics, rates and causes, greater sample sizes, clearer inclusion and exclusion criteria, and studies that measure interventions against control groups as standard. The review also emphasized the need for patient acceptability ratings.

Eight years on from the publication of the original review, and it would appear that little has changed within this research field. The authors of this update decided to use a tighter inclusion criteria than used previously to ensure only studies that measured body image or appearance-related distress were included within the analysis. This limited the number of new studies to just four. This is of fundamental importance. A recent review of self-help literature (Muftin & Thompson, 2013) included all interventions that focused on any condition that may be considered a visible difference. These studies employed various outcome measures, but many did not include a direct measure of appearance-related distress or body image. As this is the key variable in measuring level of distress in this population (Clarke et al., 2008), it is important that future interventions measure this factor as their primary outcome.

Furthermore, of the four new studies included in this update, only one consisted of a RCT reported in sufficient detail for low risk of bias and suitable for data pooling (Bessell et al., 2012). As this study was conducted by the two of the authors of this review, it demonstrates how important this timely update is for reminding future researchers of the importance of rigorous experimental design.

Current practice involves very limited testing of the efficacy of interventions, and this needs to be addressed. In the modern health arena increasing focus is being placed on evidence-based practice (Anderson, 2006). It is important that future research provides the necessary level of rigour in order to establish such evidence-based practice to ensure clinicians can work with interventions known to be effective for the target population.

Strengths and Limitations of this study

Credit must be given to the existing studies for trying to evaluate interventions for such a hard-to-reach population. Designing interventions specifically for certain conditions classified as affecting appearance can be very difficult due to the rarity of some conditions. Even when designing interventions for a wide range of conditions, the population can still be difficult to reach leading to low sample sizes and the population can vary widely, making generalizability a problem. Therefore, this review was based on small populations and meta-analysis was not possible due to differences in study design. Future research needs to consider the use of multi-site studies in order to recruit larger numbers of participants and thus increase the reliability of the findings of such evaluations.

Implications for policy and practice

It must be emphasized that despite the methodological problems associated with assessing these interventions, the techniques themselves are still important. There is minimal service provision for individuals with visible differences (Bessell & Moss, 2007) and yet there is a large population in need. It is estimated that over one million in the United Kingdom alone have some form of visible difference (Rumsey & Harcourt, 2012), and of these approximately over one third experience moderate to severe levels of distress (Rumsey et al., 2004). Although their efficacy still needs further establishment, these interventions are necessary for increasing service provision for individuals with visible differences. They are also needed to address the issue of an overall package of care for visibly different clients from medical treatment right through to adjustment and psychosocial functioning. For these reasons, further testing of these interventions is a fundamental step.

The current interventions have provided very limited support for the CBT and combined CBT and SST models. These techniques offer individuals practical solutions to some of their social difficulties without pathologising them. It is important that clinicians look to the small number of RCTs that have been conducted in this area to guide their choice of therapeutic approach with clients with visible differences. Currently, these RCTs consist of ones that assess CBT (Newell & Clarke, 2000; Papadopoulos, Walker & Anthis, 2004) and those that assess a combined social skills and CBT approach (Bessell et al., 2012). This is in keeping with the literature for body image issues in the general population, where CBT-based approaches have been found to be most effective (Jarry & Ip, 2005).

Furthermore, evidence from the acceptability measures used in some of the studies that involved these approaches has suggested that individuals with visible differences do find these types of interventions acceptable (Bessell et al., 2012; Newell & Clarke, 2000). This is further supported by a felt needs assessment recently conducted with potential service users within the field of visible difference, which identified that most service users found the idea of CBT or SST to be acceptable and positive (Bessell et al., 2010). This is an interesting point to note, as it demonstrates that individuals with visible differences do not find the idea of interventions associated with their appearance stigmatizing, as has often been a concern by experts in the past.

Conclusion

Overall, this review concludes that to date there is very limited evidence to support the efficacy of CBT or a combined CBT and SST approach for supporting adults with visible differences. However, there is still insufficient information to draw firm conclusions and little to no information available regarding the optimal setting for interventions of this nature, the optimal service provider, length of time or intensity of intervention. These are all areas that future research needs to address. All these factors must be addressed in order to demonstrate efficacy in the future. Furthermore, researchers and clinicians alike must ensure key outcome measures are included when administering or testing psychosocial interventions. In this instance, some measure of body image or appearance-related distress is important to ensure that interventions are specifically targeting appearance-related factors. The authors conclude that little has changed in the research community since the publication of the initial review. It is important that future research follows the recommendations made within these reviews in order to provide strong evidence-based practice recommendations for clinicians in the future.

Supplemental Information

Appendix Appendix

Appendix A: sample search strategy

Click here for additional data file.

Supplemental Information 2 PRISMA checklist

Click here for additional data file.

Thanks to Julie Griffin, Natasha-Ann Deeprose, Hannah Wadland, Amir Massoumian and Jo-Lee Hankinson for support with title and abstract checking and risk of bias assessment.

Additional Information and Declarations

Competing Interests

Author Contributions

1 The aim of this review was specifically to assess appearance-related distress not general psychosocial functioning and differs to other reviews, e.g., (Muftin & Thompson, 2013).

The authors declare there are no competing interests.

Alyson Norman conceived and designed the experiments, performed the experiments, analyzed the data, contributed reagents/materials/analysis tools, wrote the paper, prepared figures and/or tables.

Timothy P. Moss conceived and designed the experiments, performed the experiments, contributed reagents/materials/analysis tools, wrote the paper, reviewed drafts of the paper.

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
