# Peer review of "Psychosocial interventions for adults with visible differences: a systematic review"

_PeerJ, doi:10.7717/peerj.870_

## Round 0.1 · original submission · Major Revisions

Please note the comments of both reviewers that you must refocus this review given that your only result was that the review as written provides nothing new! Surely, you can change your focus so that you get a review of an important new area in this research arena.

Reviewer 1 ·

Basic reporting

1. too verbose. Needs at least 30-50% reduction in size
2. Unfocused. Too many conditions, too many types of interventions, too few studies
3. Poor quality of studies
4. Essentially it is garbage-in garbage-out and report of no knowledge. Ok but this does not justify such a lengthy paper just to say that 'ok there is nothing out there'
5. Accept as a brief report only

Experimental design

1. Unfocused. Too many conditions, too many types of interventions
2. otherwise ok

Validity of the findings

garbage in garbage out and this is more or less awknowledged by the authors

Additional comments

Accept as a brief report only. Essentially we knew already

·

Basic reporting

Whilst this is in general an important area for study the rationale for the basic assumptions/definitions and inclusion criteria used is not entirely clear (For example: "It was considered that the needs of individuals with “hidden” differences might be different to those with normally visible differences, meaning that different intervention techniques may be appropriate" This assumption would benefit further explanation especially as this is one of the ways many studies were apparently excluded)
The reference in the discussion to specific interventions that are not specifically referred to in the papers evaluated is hard to understand and the reader can evaluate why these interventions matter as much as the authors claim ("interventions are necessary for increasing service provision for individuals with visible differences. These include… ")

Experimental design

This was a simple qualitative literature review conducted in reasonable manner. The rationale for the choices made could have been clearer.

Validity of the findings

By their own admission, the authors admit this paper has little to contribute over and above a recent review of the same subject (“The findings of this review were no different to the conclusions of the original review”) and the overall conclusion (that more research is needed) is vague and not detailed enough to support a discussion of what the future direction of research should be. In addition no real light is thrown on what specific questions further research should ask or where in the wider literature clinicians might look for guidance. The authors speculate on issues of funding and service priorities but this is not directly supported by any of the preceding analysis of the literature. There was very little new to learn from this paper. The speculation in the discussion, whilst not entirely irrelevant or uninteresting is somewhat disconnected from the results of the actual literature review and the otherwise negative finding (For example: “The authors suggest that the reason for the lack of scientifically tested interventions is that many self-funded charities have had to pick up the shortfall in service provision and these organizations have been more concerned with spending money on providing services than on evaluating them” This is speculation and it is not at all clear how this follows from the previously detailed negative findings).

Additional comments

A narrower question concerning the impact of interventions on visible difference drawn needs to be formulated and addressed by a review of the wider literature which might then make for some original discussion. Unfortunately as it stands the paper yields little of use for clinician readers.

---

## Round 0.2 · accepted · Accept

Thanks for your response to the reviewers.